# Long-Term Influence of the Practice of Physical Activity on the Self-Perceived Quality of Life of Women with Breast Cancer: A Randomized Controlled Trial

**DOI:** 10.3390/ijerph17144986

**Published:** 2020-07-10

**Authors:** Jose L. García-Soidán, Ignacio Pérez-Ribao, Raquel Leirós-Rodríguez, Anxela Soto-Rodríguez

**Affiliations:** 1Faculty of Education and Sport Sciences, University of Vigo, Campus a Xunqueira, s/n. 36005 Pontevedra, Spain; jlsoidan@uvigo.es (J.L.G.-S.); grupomovetesl@gmail.com (I.P.-R.); 2Faculty of Physical Therapy, University of Vigo, Campus a Xunqueira, s/n. 36005 Pontevedra, Spain; 3Health Service from Galicia (SERGAS), Galician Health Services—Ourense Hospital, 32005 Ourense, Spain; anxelasoro@hotmail.com

**Keywords:** breast neoplasms, survival, exercise, quality of life, aged, women

## Abstract

*Background:* There is still no consensus on the most suitable interventions for exercise practice in breast cancer survivors. Therefore, the aim of this study was to evaluate the effect of a two-year physical activity intervention (strength, aqua fitness and aerobic exercise programs) on the self-perceived quality of life and physical functionality of female breast cancer survivors. *Methods:* A randomized, controlled, experimental trial with a sample of 316 women (63 ± 7 years), who had been diagnosed with breast cancer. The evaluations were performed using the Rikli & Jones Senior Fitness Test, and the Short Form 12 Health Survey (SF-12). *Results:* The participants in the strength program showed statistically significant improvements in all the items of the SF-12. The aqua fitness program obtained significant improvements in Physical Functioning and Limitations, Pain and Emotional Limitations, General Health, Vitality, Social Functioning and the physical and mental components of the SF-12. The participants in the aerobic program showed a progressive deterioration of Vitality and Mental Health. *Conclusion:* When assigning breast cancer survivors to an exercise program, the preferential or predominant activity should include strength exercises. On the other hand, as the second choice, those patients with particularly low levels of Vitality or Physical Limitations will show greater improvement with an aqua fitness program.

## 1. Introduction

Breast cancer is among the main causes of morbidity and mortality all over the world, with approximately 14 million new cases and more than 8 million deaths each year [1]. In the female population, the most prevalent forms of cancer are breast, colorectal, lung, cervical and stomach cancer [2]. The relative survival 5 years after diagnosis is above 80% in developed countries, which, in turn, are also the ones that implement systematized mammography-screening programs [3].

Although the aim of reaching 100% survival in women with breast cancer is still being pursued [4], improving the quality of life and restoring the optimal functionality levels of the patients are also objects of study nowadays [5,6].

The benefits of practicing physical activity (PA) on the general population at the physical and emotional level have been extensively studied [7,8], and cancer patients can also benefit from all those positive health effects [6]. At the physical level, PA practice in cancer survivors facilitates the recovery of the previous functional capacity, strength and flexibility levels, the healthy parameters of body composition, as well as the reduction of neutropenia, anemia, thrombocytopenia, pain and fatigue (the latter five are frequent side effects of aggressive cancer treatments) [9,10]. Based on these benefits and recommendations, there is value to the efforts being made to connect breast cancer survivors to high-quality strength training programs [11]. Traditionally, more research has been done on the effects of aerobic exercise on patients with malignant disease. That said, more attention is currently being paid to the effects of other training modalities (such as strength training or aqua fitness) on the physical work capacity of cancer patients or survivors. In any case, PA programs should be evaluated and implemented for their positive effects on muscle atrophy induced by the treatments and sedentary habits of breast cancer survivors [12]. Aerobic training protocols involve short periods of exercise at a vigorous intensity, followed by brief, low intensity recovery breaks, which permit the relief from symptoms such as dyspnea and leg fatigue. Aerobic programs have been tolerated in a wide range of patient groups, including individuals with chronic obstructive pulmonary disease, metabolic syndrome, heart failure and obesity. Furthermore, the aerobic interval exercise programs are safe, and caused low levels of cardiac stress [13]. However, some particularities must be taken into account when designing interventions aimed at cancer patients, such as avoiding movements that cause pain, sudden or big changes in blood pressure and heart rate, and the reaching of high levels of dyspnea [5].

In addition to the benefits for the physical clinical parameters, it is important to highlight that, since the diagnosis of cancer represents a situation of considerable emotional stress, the benefits of PA practice at the psychological level (such as the reduction of depression levels and the improvement of strength, self-esteem and emotional control) are of special interest in cancer patients [14,15].

However, there is still no consensus on the most suitable interventions and specific training parameters for PA practice in breast cancer survivors. Therefore, the aim of this study was to evaluate the effect of a two-year PA practice program on the self-perceived quality of life and physical functionality of female breast cancer survivors, and to determine the existence of differences between three different interventions (strength, aqua fitness and aerobic exercise programs).

## 2. Materials and Methods

### 2.1. Experimental Design and Sample

This was a randomized, controlled, experimental trial with a 3 × 3 design. The sample consisted of 316 women (mean age: 63 ± 7 years) from the region of Ourense (Spain), who had been diagnosed with breast cancer, surgically treated, and subjected to chemotherapy, which they had completed in the previous six months. The selection criteria excluded those patients who had heart or coronary diseases that contraindicated the practice of exercise, hypertension, severe anemia, risk of fracture, disabling osteoarticular pathology, diabetes or other disabling diseases. 

These were distributed in four groups of activity as shown in the flow diagram of Figure 1. For this research, a program for the promotion of PA and health was designed for women with cancer promoted through informative posters placed in different social centers and public boards (throughout the six months prior to the intervention), distributed by all the districts of the city. Participation was voluntary throughout the program, and distribution in all three physical activity groups and the control group was performed randomly among the participants. There was an initial medical evaluation confirming that all participants were able to participate in the intervention program.

The evaluators that recorded data in the different evaluation sessions did not know to which intervention group the participants belonged. Furthermore, during the statistical analysis, the authors who calculated the results also did not know how the intervention groups were categorized.

Patients who met the inclusion criteria were randomly assigned to one of four groups. Blank folders were numbered from 1 to 316, and were given concealed codes for the assigned group, determined by a random-number generator. When a patient was eligible and gave consent to participate, the treating monitor drew the next folder from the file, which determined the group of assignment.

All of them signed informed consent in accordance with the Declaration of Helsinki (rev. 2013) and the Data Protection Act 15/1999. This research received ethical approval from the Commission of Ethics of the Faculty of Sciences of Education and Sport of the University of Vigo (Pontevedra, Spain) (code: 3-0504-16).

### 2.2. Procedure

The evaluations were performed using the Rikli & Jones Senior Fitness Test [16,17], which evaluated functional fitness, and the Short Form 12 Health Survey (SF-12) Questionnaire [18,19,20] that measured quality of life in relation to health. The strength and flexibility of the upper and lower extremities were extracted from the Rikli & Jones Senior Fitness Test as study variables (strength is quantified according to the number of repetitions and flexibility is quantified in centimeters, through four simple tests) [16,17]. On the other hand, the following variables were extracted from the SF-12 Questionnaire: the pain, the vitality, the physical and social functions, the general and mental health; the physical and emotional limitations; and two global scores on the physical and mental components of health (all the variables provided by this questionnaire are quantified in score points) [18,19,20].

During the investigation, three data points were taken: (a) Pre-test, prior to the start of the program, a doctor checked the health status of the participants and the possible limitations or contraindications to their performance of any of the program’s activities. Then the anthropometric measures of height, weight and body mass index (BMI) were taken. Finally, the Rikli & Jones Senior Fitness Test (SFT) and the SF-12 Questionnaire were applied to the participants; (b) Intermediate test, at 12 months of onset (the same measurements were taken as at the beginning, except for the medical interview); (c) Post-test, at 24 months from the beginning, with identical parameters to those employed in the intermediate test. All evaluations were performed by qualified professionals with training in these procedures, who did not know which intervention group each of the participants belonged to. The participants who abandoned the study or missed over 25% of the sessions were removed from the final database.

### 2.3. Programs of PA

All participants were divided into four groups: one as control (who should not make any changes in their lifestyle, incorporating any new physical activity) and three activity groups of 79 people each. For the realization of the classes, the groups were divided into subgroups of 17–18 women. All programs consisted of two weekly sessions with two weeks of rest at Christmas, one week of rest at Easter and one month of rest in summer (August). Each year, 45 weeks of training were held and, in total, the women had to attend a minimum of 135 of the 180 sessions held within the intervention period. The different programs were given by monitors, who were graduates in PA and Sports Sciences, and were previously trained for the study.

(a) Strength group: Participants in this activity had a mean age of 63 ± 7 years. The activity sessions lasted 55–60 min. The work with gym machines had a progression adapted to the tests performed by the participants. During the first six weeks, the sessions consisted of an initial warm-up time of 10 min, performing general mobility exercises and stretching, followed by horizontal training with gym machines of 30–40 min, where the functioning of each muscle machine was progressively taught, which together formed a circuit of 8 exercises that worked the large muscle groups of the upper and lower limbs. Two sets of 12 repetitions with loads of 50–60% of the maximum resistance test (MR) were performed, and the 10 min at the end were dedicated to the stretching of the muscle groups worked during the session. From the seventh week loads increased, and maximum strength tests (MR) were performed to find out the individual work percentages. The strength program was initiated in the seventh week, at 60% of MR, then the repetitions were increased to 20 at weeks 7 and 8, and finally during the last four weeks the participants completed circuits of 3 series between 60% and 80% of the MR, with 10 repetitions per set and 2-minute breaks between each set.

(b) Aqua fitness Group: Participants in this activity had a mean age of 62 ± 6 years. The pool used had a depth of 1.4 m at the central area, and 1.75 m at the end. The classes lasted 55 min and the timing of this activity was as follows: An initial period of 2 weeks of low intensity. This initial period served to evaluate the average level of each group. They then went on to an improvement stage, where the repetitions, and then the intensity, were progressively increased between weeks 3 and 12. The basic structure of the sessions consisted of 5 initial minutes of joint mobility and warm-up outside the pool, followed by aerobic and choreographed exercises (25 min), strength-resistance work (10 min) in which the body regions varied according to the purpose of the session (e.g., chest, shoulder and dorsal region, arm and forearm region, lower limbs and muscles of the abdomen region), games (10 min), and finally stretches (5 min).

(c) Aerobic exercise group: This was formed of participants with a mean age of 64 ± 7 years. These lasted for 55 min, with a minimum warm-up time of 10 min (with choreographed basic aerobic steps focusing on mobility and short displacements), and 5 min of stretches at the end. All sessions also included a central component of 40 min in which mainly choreographed aerobic exercises (performed with a symmetric methodology) and, eventually, some strengthening exercises without loads, were performed (2 sets of 12 repetitions for each large muscle groups of the upper and lower limbs).

### 2.4. Data Analysis

The statistical package SPSS (version 22, SPSS Inc., Chicago, IL, USA) was used for the treatment of the data. The variables showed a normal distribution according to the Kolgomorov–Smirnov test (*p* > 0.05), and there was homogeneity of variances, determined by applying the Levene test. A factorial, repeated measures ANOVA compared the evolution of each activity group throughout the program, and to compare the effect of the different exercise programs we used the ANOVA statistic with the Bonferroni correction. The level of significance was set at *p* < 0.05.

## 3. Results

### 3.1. Strength Program

The participants in the strength program showed statistically significant improvements in all the items of the SF-12 (Table 1), except in Vitality, in which they obtained a significant worsening at the end of the intervention (*p* < 0.05). The results of Social Functioning, Physical and Emotional Limitations and Pain showed a progressive improvement (*p* < 0.001), which was more pronounced after the second year of the intervention (Figure 2). Physical Functioning obtained better results at the end of the study, although the benefits in this aspect were greater after the first year of the intervention (*p* < 0.05). Lastly, Global and Mental Health, and the physical and mental components of the SF-12, showed worse results after the first year of the intervention; however, there were significant improvements in these parameters at the end of the intervention, with respect to the initial state of the participants (*p* < 0.05).

In the SFT, the participants obtained significantly better results relating to the strength of the lower limbs after the intervention (*p* < 0.001) (Table 1), although this parameter showed worse results after the first year of the intervention (Figure 3). The flexibility of all the extremities increased significantly (*p* < 0.001). On the other hand, no improvement was found in the strength of the upper limbs in any of the two re-evaluations. Body weight and BMI were significantly lower after the intervention (*p* < 0.01).

### 3.2. Aqua Fitness Program

The participants in the aqua fitness program obtained progressive improvements in Physical Functioning and Limitations, Pain and Emotional Limitations (*p* < 0.01) (Table 1). In General Health, Vitality, Social Functioning and the physical and mental components of the SF-12, the participants completed the intervention with significantly better results (*p* < 0.01), although they obtained worse results in all of them after the first year of the intervention (Figure 2). Lastly, there were no significant changes in Mental Health. Regarding the SFT, the participants of the aqua fitness group obtained a significant improvement in flexibility (*p* < 0.001), although their upper limb strength decreased significantly (*p* < 0.001). On the other hand, lower limb strength, body weight and BMI showed no changes after the intervention (Figure 3).

### 3.3. Aerobic Exercise Program

The participants in the aerobic exercise program showed progressive improvements in Emotional Limitations (*p* < 0.001). On the other hand, they obtained a progressive deterioration of Vitality and Mental Health (*p* < 0.01). Physical Functioning, after the intervention, showed significantly better results (*p* < 0.05) (Table 1), although such improvement was greater after the first year of the intervention (Figure 2). Physical Limitations and Pain improved after the first year of the intervention, and showed an even greater improvement after the second year (*p* < 0.05). Lastly, General Health, Social Functioning and the physical and mental components of the SF-12 obtained significantly better results after the intervention (*p* < 0.01), although they showed worse results after the first year.

Regarding the results of the SFT, the participants of the aerobic exercise program obtained a significant improvement in the strength and flexibility of their lower limbs (*p* < 0.05); however, they showed a decrease in the strength and flexibility of their upper limbs (*p* < 0.001). The body weight and BMI of these participants increased after the intervention, although not significantly (Figure 3).

### 3.4. Comparative Results between Groups

The results of the strength program group were positive in all the analyzed variables, except Vitality, which showed worse results at the end of the intervention with respect to the baseline levels. Moreover, the results of the strength group were significantly higher than those of the other two programs in body weight, BMI, Emotional Limitations, Social Functioning, Mental Health and the mental components of the SF-12. It was also the intervention that achieved the best results in the SFT battery, except for the flexibility of the lower limbs, which obtained its best improvement in the aqua fitness program.

Among the three different interventions, the aqua fitness group showed the best results in Physical Functioning, Physical Limitations, Vitality, Pain and the physical component of the SF-12. On the other hand, upper limb strength was the only variable with worse results at the end of this intervention.

Lastly, the aerobic exercise group obtained the best results in General Health. In contrast, it was the only group with worse results in body weight, BMI, Physical Limitations, Pain, Vitality, Mental Health, and upper limb flexibility and strength; in fact, the results of the last two variables after the two years of intervention were worse than those of the control group.

## 4. Discussion

The aim of this study was to evaluate the effect of a two-year PA practice program on the self-perceived quality of life and physical functioning of female breast cancer survivors, and to determine the existence of differences between different exercise-based interventions. After the analysis of the obtained data, we can conclude that the female breast cancer survivors benefit from PA practice. In addition, the design of such intervention programs has a considerable impact on the reachable benefits. 

In general, the strength program provided the best, and the most, benefits, especially in the parameters of physical health, such as body weight, BMI, general health, pain, lower limb strength and the flexibility of all extremities. These findings are congruent, since strength programs are the ones that provide the greatest benefits to those who practice them, through the development of the functional capacity (respiratory and movement capacity), the prevention of sarcopenia and the reduction of weakness, pain and other side effects derived from medical cancer treatments [21,22]. Regarding the emotional scope, the strength program induced the greatest benefits in emotional limitations and mental health, which is in line with the results of previous studies in older populations [23] and women without cancer [24], with the exception that the vitality of the participants also improved. However, it is important to take into account that, in cancer survivors, emotional aspects such as self-perceived vitality can be strongly affected by the psycho-somatic impact of the disease [25,26].

In any case, if the initial valuation of the self-perceived health state of a breast cancer survivor obtains bad results in Vitality, the results of this study indicate that the most adequate intervention is that of aqua fitness. However, in the objective valuation of the physical health state through the SFT, the aqua fitness program only obtained positive results in the flexibility of the limbs, and limb strength did not improve in any case. This may be due to the fact that aquatic exercises are especially recommended to improve fatigue levels [27] (a variable of great interest in cancer survivors [28,29]) and abilities like coordination and balance [30]. This explains the positive results obtained in this study for Physical Functioning and Limitations via the aqua fitness program: by improving fatigue and coordination, the self-perceived impact on the development of the activities of daily living may be greater than the impact identified as a consequence of the specific improvement of upper limb strength (as was the case in the strength program group).

Lastly, the intervention based on aerobic exercise obtained the worst results. This program induced a worsening in the physical limitations, vitality, mental health, pain and upper limb strength and flexibility, reaching lower values than the control group in some of these parameters. It is important to highlighting that aerobic exercise programs can improve two physical capacities: aerobic capacity and limb movement range [31,32]. Since such improvements can also be obtained with the other two PA programs, it can be confirmed that aerobic exercise programs should not be considered as the first choice when incorporating PA practice after the active treatment of breast cancer, unless the patient has a special preference for this type of physical activity.

Therefore, when assigning breast cancer survivors to a PA program, the preferential or predominant activity should include strength exercises. Moreover, if their initial results are worse as regards pain, general and mental health of the SF-12 or limb strength and flexibility, they are likely to further benefit from strength training. On the other hand, as the second choice, those patients with particularly low levels of vitality or physical limitations will show greater improvements under a PA program that includes aquatic activities, such as aqua fitness. Lastly, if these two options are not suitable for the patient, an aerobic exercise program can be considered, whose benefits on general health, emotional limitations and social functioning are greater than those obtained by the aqua fitness program. It is worth highlighting that the three interventions proved to be safe; no adverse effects were reported by the participants in any of the exercise sessions throughout the two-year period of the programs (such as lymphedema, which did not appear in any of the participants). However, it is important to mention that the mortality of the participants was significantly higher in the aqua fitness group, although this is a multi-factor phenomenon that is beyond the scope of the present investigation.

Regarding the limitations of this study, more sensitive and specific objective measurements should have been included (such as cardiovascular and laboratory tests). However, this is the first study to carry out PA interventions comparatively, using a randomized and controlled methodology for a period of time long enough to obtain reliable results that can also be extrapolated to those of other populations of female breast cancer survivors.

Considering future research lines, similar studies could be conducted in other populations to improve the evidence of the findings presented in this work, including, if possible, medical laboratory tests, to compare the self-perceived changes of the participants.

## 5. Conclusions

PA practice in female breast cancer survivors has multiple and considerable health benefits. Strength training programs obtained the best, and the most, benefits in the quality of life related to self-perceived health and general physical state. Other valid strategies for the increase of PA practice for these patients include aqua fitness programs and, as the last option, aerobic exercise programs, unless the patient has especially important afflictions pertaining to physical limitations and social functioning, in which case the practice of aerobic exercise should be the first choice.

## Figures and Tables

**Figure 1 ijerph-17-04986-f001:**
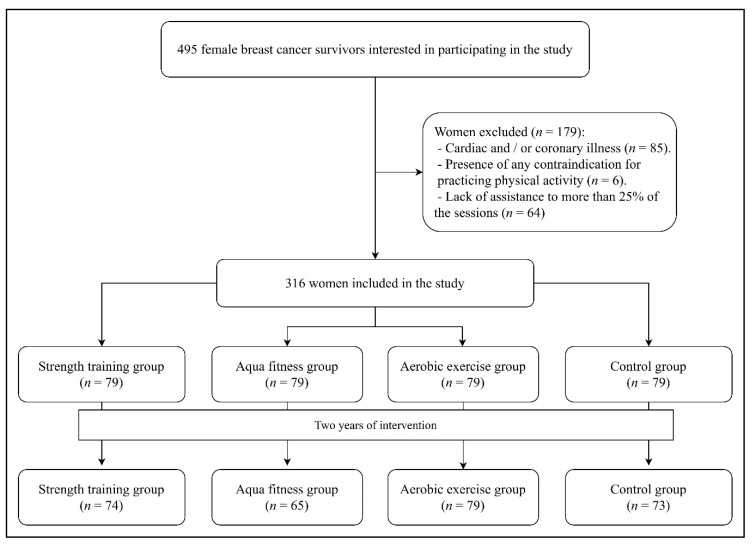
CONSORT flow diagram.

**Figure 2 ijerph-17-04986-f002:**
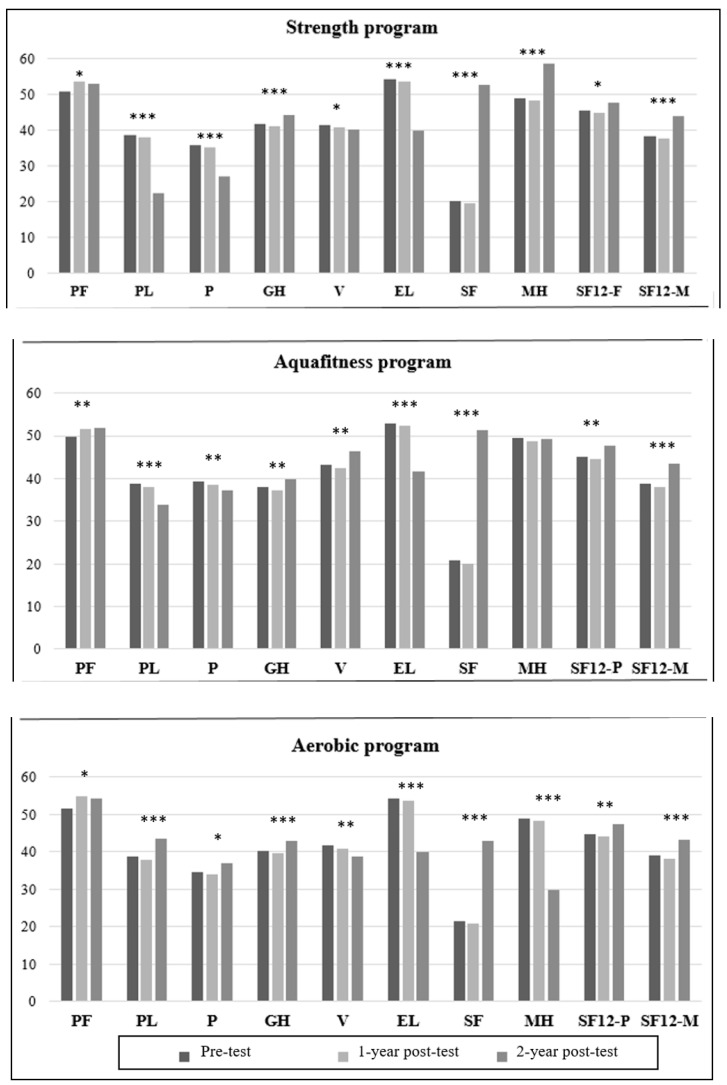
SF-12 survey results by intervention group. (PF: physical function; PL: physical limitations; P: pain; GH: general health; V: vitality; EL: emotional limitations; SF: social function; MH: mental health; SF12-P:SF12 physical component; SF12-M: SF12 mental component). Comparison between initial test vs. final test: * *p* < 0.05; ** *p* < 0.01; *** *p* < 0.0001.

**Figure 3 ijerph-17-04986-f003:**
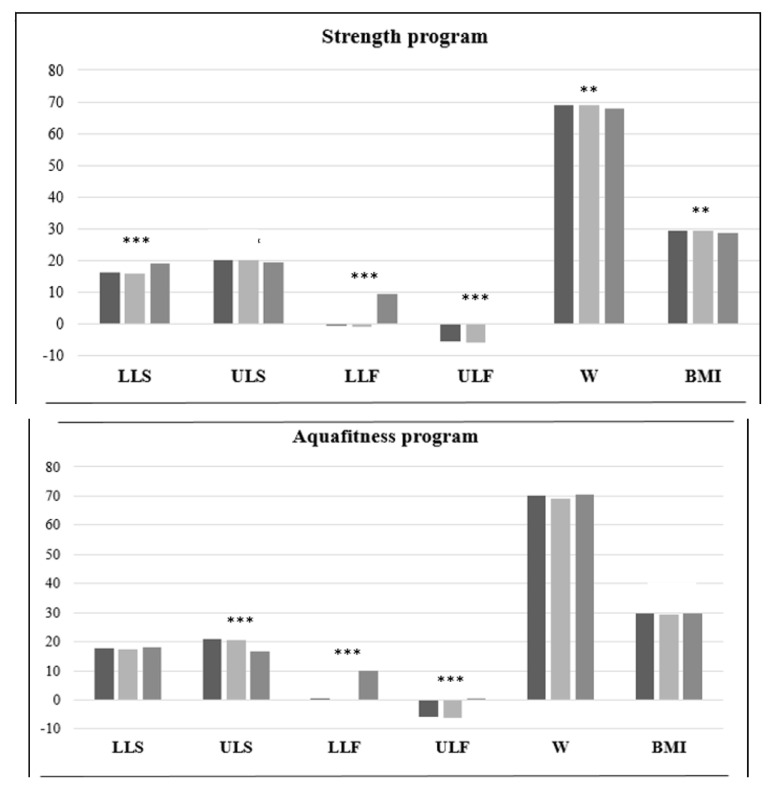
Senior Fitness Test, weight and body mass index results by intervention group. (LLS: lower limbs strength; ULS: upper limbs strength; LLF: lower limbs flexibility; ULF: upper limbs flexibility; W: weight; BMI: body mass index). Comparison between initial test vs. final test: * *p* < 0.05; ** *p* < 0.01; *** *p* < 0.0001.

**Table 1 ijerph-17-04986-t001:** Descriptive analysis of the sample by groups (mean ± standard deviation).

	Strength Group		Aqua Fitness Group		Aerobic Group		Control Group	
	Pre-Test	Post-Test	Pre-Test	Post-Test	Pre-Test	Post-Test	Pre-Test	Post-Test
Survival (*n*)	79	74	79	65	79	72	79	73
Age (years)	63 ± 7	65 ± 7	62 ± 6.8	64 ± 6.8	64 ± 7.1	66 ± 7.1	63 ± 4.6	65 ± 4.6
Weight (kg)	69 ± 11	67.7 ± 10.2 ^#**+^	70.5 ± 11	71.2 ± 10.9 ^%%**$$$^	65.7 ± 9.8	66 ± 10.5 ^&&+$$$^	69.5 ± 10.6	69.7 ± 10.6 ^#%%&&^
BMI (kg/m^2^)	29.4 ± 8.5	28.8 ± 9 ^#**+^	29.8 ± 9.6	30.2 ± 9.5 ^%%**$$$^	27.7 ± 8.3	27.9 ± 9.1 ^&&+$$$^	28.5 ± 9.4	28.6 ± 9.3 ^#%%&&^
**SF-12 health survey**								
PF	50.8 ± 9.2	53.1 ± 6.6	49.8 ± 8.1	53 ± 7.2	51.5 ± 7.7	52.7 ± 6.7	53.2 ± 9.1	53.6 ± 7
PL	38.7 ± 1.3	34.4 ± 19.7 ^#++^	38.7 ± 1.4	34 ± 21.9 ^%%%$$$^	38.7 ± 1.4	43.7 ± 23.7 ^++$$$^	38.9 ± 22.9	44 ± 21 ^#%%%^
Pain	36 ± 13.7	34.1 ± 8.4 ^##**++^	39.4 ± 13.7	37.1 ± 1.4 ^**^	34.6 ± 11.5	37.1 ± 1.4 ^++^	27.2 ± 1.3	37.3 ± 1.6 ^##^
GH	41.8 ± 9.3	44.4 ± 13.7 ^##**+^	38.1 ± 8.3	39.8 ± 13.7 ^%%%**$$^	40.3 ± 9.9	43 ± 11.5 ^&&+$$^	42.3 ± 9.3	25.6 ± 15.2 ^##%%%&&^
Vitality	41.4 ± 5.8	39.2 ± 9.3^**^	43.2 ± 12.1	46.5 ± 8.3 ^%%**$$^	41.6 ± 14.2	38.7 ± 9.9 ^&$$^	38.3 ± 12.9	40.7 ± 9.8 ^%%&^
EL	54.4 ± 6.4	39.8 ± 5.8 ^##*^	53 ± 4.1	41.6 ± 12.1 ^%%*$^	54.4 ± 10.9	40 ± 14.2 ^&&$^	54.5 ± 6.4	36.7 ± 13.1 ^##%%&&^
SF	20.2 ± 3.9	52.8 ± 6.4 ^###*^	20.7 ± 4.3	51.4 ± 4.1 ^%%%*$^	21.4 ± 4.5	52.8 ± 10.9 ^&&$^	20.9 ± 3.9	50.9 ± 7 ^###%%%&&^
MH	49.1 ± 7.8	58.6 ± 3.9 ^###***+++^	49.4 ± 7	49.1 ± 4.3 ^%%%***$$$^	48.9 ± 8.5	45.8 ± 4.5 ^&&+++$$$^	48.5 ± 7.8	49.3 ± 3.6 ^###%%%&&^
SF12-P	45.6 ± 4.2	47.5 ± 7.8 ^#^	45.1 ± 4.1	47.8 ± 7 ^%%^	44.8 ± 3.8	47.3 ± 8.5 ^&^	43.8 ± 4.5	46.9 ± 7.4 ^#%%&^
SF12-M	38.4 ± 1.4	44 ± 4.5 ^#+^	38.9 ± 4.2	43.5 ± 4.1 ^%^	39 ± 4.5	43.2 ± 3.8 ^+^	38.1 ± 5.4	42.2 ± 4.5 ^#^
**Senior Fitness Test**								
LLS	16.2 ± 0.6	18.3 ± 1	17.7 ± 4.5	18.2 ± 1.4	17.1 ± 4.9	18.8 ± 1.7 ^&^	17.6 ± 5.8	17.2 ± 1.3 ^&^
ULS	20.2 ± 7.6	19.2 ± 0.6 ^##**+++^	20.9 ± 5.5	16.7 ± 4.5 ^**^	20.2 ± 5.5	16.1 ± 4.9 ^+++^	19.6 ± 9.6	16.6 ± 4.9 ^##^
LLF	−0.7 ± 5.2	19.2 ± 7.6 ^#^	0.2 ± 8.6	19.9 ± 5.5 ^%%^	0.7 ± 8.8	19.2 ± 5.5	−0.1 ± 9	18.6 ± 5.8 ^#%%^
ULF	−5.5 ± 7.4	0 ± 9.4 ^##+++^	−5.7 ± 12	0.5 ± 10.2 ^%%$$^	−5.5 ± 13.3	−6.8 ± 8.8 ^&&+++$$^	−10.1 ± 4.9	−2 ± 9.8 ^##%%&&^

BMI: body mass index; PF: physical function; PL: physical limitations; GH: general health; EL: emotional limitations; SF: social function; MH: mental health; SF12-P: SF12 physical component; SF12-M: SF12 mental component; LLS: lower limbs strength; ULS: upper limbs strength; LLF: lower limbs flexibility; ULF: upper limbs flexibility. Comparison between strength vs. control groups: ^#^
*p* < 0.05; ^##^
*p* < 0.01; ^###^
*p* < 0.001. Comparison between aqua fitness vs control groups: ^%^
*p* < 0.05; ^%%^
*p* < 0.01; ^%%%^
*p* < 0.001. Comparison between aerobic vs control groups: ^&^
*p* < 0.05; ^&&^
*p* < 0.01; ^&&&^
*p* < 0.001. Comparison between strength vs. aqua fitness groups: * *p* < 0.05; ** *p* < 0.01; *** *p* < 0.001. Comparison between strength vs. aerobic groups: ^+^
*p* < 0.05; ^++^
*p* < 0.01; ^+++^
*p* < 0.001. Comparison between aqua fitness vs. aerobic groups: ^$^
*p* < 0.05; ^$$^
*p* < 0.01; ^$$$^
*p* < 0.001.

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
