# Peer review of "Long-Term Influence of the Practice of Physical Activity on the Self-Perceived Quality of Life of Women with Breast Cancer: A Randomized Controlled Trial"

_ijerph, 2020, doi:10.3390/ijerph17144986_

Round 1

Reviewer 1 Report

This is a very interesting and most relevant study that recruited Spanish women breast cancer survivors into a multi-intervention study contrasting three different physical activity regimens with the women then followed for up to 2 years. Three activity interventions were evaluated and a control group of women were included in the study design.

Issues

  1. The authors say that this is a double-blind randomized study. However, I do not see how one can double-blind a physical activity program. The women know what group they are in…certainly the trainers do also.
  2. There is little discussion of how women were randomized. In Table 1, there appear to be some differences between groups in various parameters, however, they are relatively similar which would suggest randomization occurred. That said there are some differences and it was not discussed if any adjustment for those differences might be relevant to consider.
  3. It is stated that women who dropped out/missed more than 25% of activity sessions were deleted. There is no discussion, however, of # women who dropped out and then the # with lower attendance. The analyses are therefore not ‘intent to treat’ but rather of women who completed the activity sessions. The loss was most notable for those women in the aquafitness activity regimen.
  4. The three activity regimens were described, although it was not clear that the intervention occurred over the entire 2 years. Might be useful to include average # sessions attended per group.
  5. Analysis methods and then results need substantially more clarity.
    1. Variables of interest need to be made clearer in the methods section. There is some description at the bottom of the tables, but that is hard to decipher .
    2. Authors state they utilized ‘t tests for related samples’; presumably that means paired t-tests of the change between visits?  That would mean, however, that those people who dropped out would be excluded from the analysis which does not seem to be happening in Table 1. If comparing mean at time 1 with mean at time 2 with different sample sizes, then not controlling for baseline values (or the change over time) and including different people in the comparisons.
    3. Table 1 does show sample sizes at baseline and year 1 (although the table needs to be labeled that way). The table does not include values for Year 2, although findings are presented in Figures 1 and 2. Presumably the sample sizes dropped more at year 2?
    4. Labelling and titles for Figures 1 and 2 are not complete. What does the y-axis of values 0-60 mean in Figure 2 and then -10-80 in Figure 3? Table 1 then gives values, but that came after the Figure 2 was presented. These figures also appear to be just descriptive of the interventions since they do not include control group. If this is the mean value at each visit, then state and include SD also in the figure.
    5. It would help to have Table 1 come before the results of the Figures since that table includes the measurement values and therefore the scale of the various outcome variables. This could also be described in the results section.
  6. There needs to be some editing of the manuscript both in spelling and use of words. Some examples:
    1. page 1, line 41 – sistemitized should be systematized or systemized or systemised
    2. page 2, line 70 –affectations should probably be afflictions or diseases
    3. page 3, line 93 and 99 should be ‘Fitness’ and not Finest
    4. page 4, line 114 –‘assimilating’ should probably be ‘simulating’
    5. Page 4 uses acronyms or abbreviations that have not been defined.

Author Response

Dear Editor and Reviewer of the International Journal of Environmental Research and Public Health:

Thank you very much for your suggestions and contributions to improve the quality of the manuscript. Following your indications, we respond, point by point, to your comments.

In the text, all the modified or added sentences have been written in red to facilitate the correction by the reviewers.

  1. The authors say that this is a double-blind randomized study. However, I do not see how one can double-blind a physical activity program. The women know what group they are in…certainly the trainers do also.

The evaluators who recorded the data in the different evaluation sessions did not know which intervention group the participants belonged to. Furthermore, during the statistical analysis which intervention the participants had received, it was also not known by the authors to calculate the results.

This information has been added in the subsection Experimental design and sample.

  1. There is little discussion of how women were randomized. In Table 1, there appear to be some differences between groups in various parameters, however, they are relatively similar which would suggest randomization occurred. That said there are some differences and it was not discussed if any adjustment for those differences might be relevant to consider.

Patients who met the inclusion criteria were randomly assigned to one of four groups. Blank folders were numbered from 1 to 316 and were given concealed codes for the group of assignment, determined by a random-number generator. When a patient was eligible and gave consent to participate, the treating monitor drew the next folder from the file, which determined the group of assignment.

This description of the randomization method has been added to the manuscript in subsection Experimental design and sample.

  1. It is stated that women who dropped out/missed more than 25% of activity sessions were deleted. There is no discussion, however, of # women who dropped out and then the # with lower attendance. The analyses are therefore not ‘intent to treat’ but rather of women who completed the activity sessions. The loss was most notable for those women in the aquafitness activity regimen.

Indeed, the group that suffered the most losses was Aqua fitness Group. In all the groups, it was found that all the women who had been leaving the program were also those who died. Although we did not establish a relationship between survival and the assigned activity group, the authors do reflect this in Table 1. This phenomenon is mentioned in Results and Discussion but without falling into speculation or statements that are beyond the scope of this research.

  1. The three activity regimens were described, although it was not clear that the intervention occurred over the entire 2 years. Might be useful to include average # sessions attended per group.

All programs consisted of two weekly sessions with two weeks of rest at Christmas, one week of rest at Easter and one month of rest in summer (August). Each year, 45 weeks of training were held and, in total, the women had to attend a minimum of 135 of the 180 sessions held within the intervention period.

This information has been added to the manuscript in subsection Programs of Physical Activity.

  1. Variables of interest need to be made clearer in the methods section. There is some description at the bottom of the tables, but that is hard to decipher.

The Procedure subsection has been expanded with the information requested:

“The evaluations were performed using the Rikli & Jones Senior Fitness Test [13, 14], which evaluated functional fitness, and the SF-12 Questionnaire [15-17] that measured quality of life in relation to health. The strength and flexibility of the upper and lower extremities were extracted from the Rikli & Jones Senior Fitness Test as study variables (strength is quantified according to the number of repetitions and flexibility is quantified in centimeters, through four simple tests) [13, 14]. On the other hand, the following variables were extracted from the SF-12 Questionnaire: the pain, the vitality, the physical and social functions, the general and mental health; the physical and emotional limitations; and two global scores on the physical and mental components of health (all the variables provided by this questionnaire are quantified in score points) [15-17].”

  1. Authors state they utilized ‘t tests for related samples’; presumably that means paired t-tests of the change between visits? That would mean, however, that those people who dropped out would be excluded from the analysis which does not seem to be happening in Table 1. If comparing mean at time 1 with mean at time 2 with different sample sizes, then not controlling for baseline values (or the change over time) and including different people in the comparisons.

The comparisons in the study variables have been made taking into account only the participants who completed the intervention with the aforementioned attendance requirements (less than 25% absenteeism from the total exercise sessions).

Regardless, in Table 1 the authors wanted to reflect the survival of the sample over the two years of the intervention. But this does not mean that the results obtained by the participants who did not finish the intervention are included in the indicated pre-test scores.

  1. Table 1 does show sample sizes at baseline and year 1 (although the table needs to be labeled that way). The table does not include values for Year 2, although findings are presented in Figures 1 and 2. Presumably the sample sizes dropped more at year 2?

The post-test scores in Table 1 belong to the final evaluation (after two years of intervention). The authors have detected the phrase of Results that gave rise to the interpretation error and we have modified it:

“In the SFT, the participants obtained significantly better results in the strength of the lower limbs after the intervention (p<0.001) (Table 1), although this parameter showed worse results after the first year of the intervention (Figure 3).”

  1. Labelling and titles for Figures 1 and 2 are not complete. What does the y-axis of values 0-60 mean in Figure 2 and then -10-80 in Figure 3? Table 1 then gives values, but that came after the Figure 2 was presented.

To make the Figures and the Table more clearly interpretable, in the Procedure sub-section the description of the assessment instruments has been expanded with their units of measurement:

“The evaluations were performed using the Rikli & Jones Senior Fitness Test [13, 14], which evaluated functional fitness, and the SF-12 Questionnaire [15-17] that measured quality of life in relation to health. The strength and flexibility of the upper and lower extremities were extracted from the Rikli & Jones Senior Fitness Test as study variables (strength is quantified according to the number of repetitions and flexibility is quantified in centimeters, through four simple tests) [13, 14]. On the other hand, the following variables were extracted from the SF-12 Questionnaire: the pain, the vitality, the physical and social functions, the general and mental health; the physical and emotional limitations; and two global scores on the physical and mental components of health (all the variables provided by this questionnaire are quantified in score points) [15-17].”

  1. It would help to have Table 1 come before the results of the Figures since that table includes the measurement values and therefore the scale of the various outcome variables.

The order of the Table and Figures has been changed. In addition, as indicated in the previous correction, the units of measurement for each study variable are now explained in subsection Procedure.

  1. There needs to be some editing of the manuscript both in spelling and use of words. Some examples: page 1, line 41 – sistemitized should be systematized or systemized or systemized; page 2, line 70 –affectations should probably be afflictions or diseases; page 3, line 93 and 99 should be ‘Fitness’ and not Finest; page 4, line 114 –‘assimilating’ should probably be ‘simulating’; page 4 uses acronyms or abbreviations that have not been defined.

The indicated errors have been corrected and the English language and style of the manuscript has been reviewed and corrected by a native English-speaking translator.

Once again, thank you very much for the time spent and the interest shown in this work; as well as in the positive evaluations you have given of it.

Receive a warm greeting,

The authors.

Reviewer 2 Report

The authors should be congratulated as they address a topic of high relevance to public health such as guaranteeing an optimal quality of life for breast cancer survivors. In addition, they use several physical conditioning trends as interventions, which are within the reach of the study population and are increasingly used to create a physical activity routine with healthy connotations. However, some major aspects must be addressed to increase the quality of the manuscript and are detailed below:

Introduction

Lines 61-62

  • It would be interesting if the authors justified why they choose these types of activities. Have these activities been tested before in patients with breast cancer? Please add more information.

Material and Methods

  • At the beginning of this section, the authors described that this study is a “Double-blind, randomized, controlled experimental study”. Why this study is double-blind if participants know the intervention received?. Please clarify and add more information about the design.
  • More information about the randomization process is needed in order to add more quality to the study.
  • More information is need about the selection process, for example:
  • How many people are the real population affected by breast cancer? It is necessary to add the percentage who agreed to participate from the real population.
  • How long time the posters had exposed in the different assistance centers?

  • In figure 1, n= 64 were excluded because of the lack of assistance > 25% of the total sessions, however, at the beginning of selection, you do not know the lack of assistance. Maybe this information must be at the end of the flow chart?. Otherwise, please be clear, please.
  • In line 103, the authors describe this expression “who knew which intervention group each of the participants belonged to”. The ideal is that they do not know which intervention the participants belong to, isn't it?. Please, clarify.
  • Please, describe the qualification of trainers or monitors.
  • Regarding the strength intervention, please indicate just at the beginning that the intervention is based on full-body strength workout using gym machines, I think that this a better concept since being honest, I can' t imagen that breast cancer survivor needs to do bodybuilding to improve her physical fitness. Bodybuilding is more related to sculpt your body for competition. Additionally, this competition reaches limits which are not necessary for this population.
  • Please, clarify the type of training, is vertical training (1x12 of each exercise in circuit x 2 rounds) or horizontal training (2x12 of each exercise). It is confused because you said that it is a circuit training but after you explain the volume of exercises as 2x12). After, the rest is between rounds of each 12 repetitions or after the finish of each circuit round?.
  • How the authors calculated the 10RM at the beginning?
  • How the authors controlled the intensity of the training?
  • Line 122, what it is MRI?
  • Aqua fitness intervention needs more details. What do the authors want to say with low intensity? How the authors quantify the intensity? How do the authors evaluate the physical level of the participants to increase the intensity of the classes? The warm-up was outside the swimming pool or inside? Had the water a specific temperature? The aerobic part was choreographed or not? There was musical support during the activity? Which type of materials were used for the strength part of the session? In this part, how the volume of the training was controlled, repetitions, time? How the intensity of the sessions was registered and controlled?.
  • Aerobic exercise intervention, also need more information. Usually, the warm-up of aerobics classes was included in the choreographed part, with basis aerobic steps focus on mobility and short displacements, please described with more details the warm-up. Regarding the main part, please add if the instructor performed a symmetric or asymmetric choreographic methodology or both. In addition, it is essential that authors indicate the bpm of the music for each part, and how they controlled the cardiovascular intensity.
  • Regarding, eventually strengthening part, please include the time of this part, typical exercises, and methodology to control the volume. I can imagine that it was controlled by rounds and repetitions based on the bpm of the music.

Please use not loads instead of autoloads.

  • Please, it is necessary that the test and questionnaire used were described with more detail in order to understand the variables analyzed in the results section (ie; dimensions of the quality of life, score, evaluation, ….). Add information about reference, validity, and other information that describes that the instrument and test used are appropriated for this population and the main outcomes of the study.
  • Some strategies to guarantee adherence were performed during the interventions?

Statistics analysis:

  • The study has a 3x3 design (please include at the beginning of the methods part), i.e.: 3 groups and 3 times of measures for that reason I think that t-test is not the correct analysis since t-test is used to analyze the mean difference between maximum two groups or two times of measures. I suggest to authors perform a factorial repeated measures ANOVA because it is the ideal statistical test for the design of this study.
  • Authors controlled other confounders?

Results.

  • Please, use subsections to describe the results according to the activity groups, i.e. describe the results within the groups during the three assessments in separated subsections.
  • Please, report the adherence or % of participants' attendance.

Discussion.

In the first paragraph of the discussion, the authors describe an expression about the rest of the population. Please, describe only your main findings related to your study sample.

In general, the discussion needs to depth in the clinical data. Please, describe with more detail the results of the mentioned studies compared with your results (i.e. quantify and compare the improvements or detriments of the outcomes between the studies in the literature and your study).

Author Response

Dear Editor and Reviewer of the International Journal of Environmental Research and Public Health:

Thank you very much for your suggestions and contributions to improve the quality of the manuscript. Following your indications, we respond, point by point, to your comments.

In the text, all the modified or added sentences have been written in red to facilitate the correction by the reviewers.

  1. Introduction (lines 61-62): It would be interesting if the authors justified why they choose these types of activities. Have these activities been tested before in patients with breast cancer? Please add more information.

This information has been added:

“The benefits of practising physical activity (PA) on the general population at the physical and emotional level have been extensively studied [7, 8], and cancer patients can also benefit from all those positive health effects [6]. At the physical level, PA practice in cancer survivors facilitates the recovery of the previous functional capacity, strength and flexibility levels, healthy parameters of body composition, as well as the reduction of neutropenia, anemia, thrombocytopenia, pain and fatigue (the latter five are frequent side effects of the aggressive cancer treatments) [9, 10]. Based on these benefits and recommendations, there is value to efforts to connect breast cancer survivors to high-quality strength training programs [11]. Traditionally, more research has been done on the effects of aerobic exercise in patients with malignant disease. But, more attention is currently being paid to the effects of other training modalities (such as strength training or aqua fitness) on the physical work capacity of cancer patients or survivors. In any case, PA programs should be evaluated and implemented for their positive effects on muscle atrophy induced by the treatments and the sedentary habits in breast cancer survivors [12].  Aerobic training protocols involve short of exercise at a vigorous intensity, followed by brief, low intensity recovery breaks, which permit the relief from symptoms such as dyspnea and leg fatigue. Aerobic programs have been tolerated in a wide range of patient groups including individuals with chronic obstructive pulmonary disease, metabolic syndrome, heart failure, and obesity. Furthermore, the aerobic interval exercise programs are safe, and caused low levels of cardiac stress [13]. However, some particularities must be taken into account when designing interventions aimed at cancer patients, such as avoiding movements that cause pain, sudden or big changes in blood pressure and heart rate, and reaching high levels of dyspnea [5].”

  1. Material and Methods: At the beginning of this section, the authors described that this study is a “Double-blind, randomized, controlled experimental study”. Why this study is double-blind if participants know the intervention received? Please clarify and add more information about the design. More information about the randomization process is needed in order to add more quality to the study.

The evaluators who recorded the data in the different evaluation sessions did not know which intervention group the participants belonged to. Furthermore, during the statistical analysis which intervention the participants had received, it was also not known by the authors to calculate the results.

Patients who met the inclusion criteria were randomly assigned to one of four groups. Blank folders were numbered from 1 to 316 and were given concealed codes for the group of assignment, determined by a random-number generator. When a patient was eligible and gave consent to participate, the treating monitor drew the next folder from the file, which determined the group of assignment.

These descriptions of the methods used have been added to the manuscript in subsection Experimental design and sample.

  1. Material and Methods: More information is need about the selection process, for example: How long time the posters had exposed in the different assistance centers?

This detail about the timing has been added:

“For this research, a program for the promotion of PA and health was designed for women with cancer promoted through informative posters placed in different social centers and public boards (throughout the six months prior to the intervention) distributed by all the districts of the city.”

  1. Material and Methods: In Figure 1, n=64 were excluded because of the lack of assistance > 25% of the total sessions, however, at the beginning of selection, you do not know the lack of assistance. Maybe this information must be at the end of the flow chart? Otherwise, please be clear, please.

Since compliance with the minimum attendance rate was considered an inclusion criteria, its position in Figure 1 is correct. However, it is an aspect that became known throughout the intervention (according to the participants they reached the maximum number of absent sessions), so this aspect has been reformulated in the figure: "Lack of assistance to more than 25% of the sessions throughout the intervention (n = 64)".

  1. Material and Methods: In line 103, the authors describe this expression “who knew which intervention group each of the participants belonged to”. The ideal is that they do not know which intervention the participants belong to, isn't it? Please, clarify.

This is a writing error, obviously the authors wanted to emphasize that the evaluators did NOT know which group the participants they evaluated belonged to. This error has already been corrected.

  1. Material and Methods: Please, describe the qualification of trainers or monitors.

This detail has been added: “The different programs were given by monitors graduates in PA and Sports Sciences and were previously trained for the study”.

  1. Material and Methods: Regarding the strength intervention, please indicate just at the beginning that the intervention is based on full-body strength workout using gym machines, I think that this a better concept since being honest, I can' t imagen that breast cancer survivor needs to do bodybuilding to improve her physical fitness. Bodybuilding is more related to sculpt your body for competition. Additionally, this competition reaches limits which are not necessary for this population.

The error that indicates us has been corrected with the use of more appropriate and specific terms.

  1. Material and Methods: Please, clarify the type of training, is vertical training (1x12 of each exercise in circuit x 2 rounds) or horizontal training (2x12 of each exercise). It is confused because you said that it is a circuit training but after you explain the volume of exercises as 2x12).

This detail has been added: The type of training was horizontal.

  1. Material and Methods: Aqua fitness intervention needs more details. The warm-up was outside the swimming pool or inside? Had the water a specific temperature? The aerobic part was choreographed or not? There was musical support during the activity? Which type of materials were used for the strength part of the session? In this part, how the volume of the training was controlled, repetitions, time?

All these details, if used, have been added to the manuscript. However, the water did not have a specific temperature, no music was used during classes, and no additional material was used for strength work (and therefore has not been indicated in the text). The volume of training was increasing according to the tolerance of the patients and always respecting any negative or fatigue signs that they manifested.

  1. Material and Methods: Aerobic exercise intervention, also need more information. Usually, the warm-up of aerobics classes was included in the choreographed part, with basis aerobic steps focus on mobility and short displacements, please described with more details the warm-up. Regarding the main part, please add if the instructor performed a symmetric or asymmetric choreographic methodology or both. In addition, it is essential that authors indicate the bpm of the music for each part, and how they controlled the cardiovascular intensity. Regarding, eventually strengthening part, please include the time of this part, typical exercises, and methodology to control the volume. I can imagine that it was controlled by rounds and repetitions based on the bpm of the music. Please use not loads instead of autoloads.

All these details, if used, have been added to the manuscript:

“(c) Aerobic exercise group: It was formed by participants with a mean age of 64 ± 7 years. These lasted for 55 minutes, with a minimum warm-up time of 10 minutes (with choreographed basis aerobic steps focus on mobility and short displacements), and 5' stretches at the end. All sessions also included a central component of 40 minutes in which mainly choreographed aerobic exercises (performed with a symmetric methodology) and, eventually, some strengthening exercises without loads were performed (2 sets of 12 repetitions for each large muscle groups of upper and lower limbs).”

But the bpm of the music was not registered.

  1. Material and Methods: Please, it is necessary that the test and questionnaire used were described with more detail in order to understand the variables analyzed in the results section (ie; dimensions of the quality of life, score, evaluation, ...). Add information about reference, and other information that describes that the instrument and test used are appropriated for this population and the main outcomes of the study.

The Procedure subsection has been expanded upon your request:

“The evaluations were performed using the Rikli & Jones Senior Fitness Test [16, 17], which evaluated functional fitness, and the SF-12 Questionnaire [18-20] that measured quality of life in relation to health. The strength and flexibility of the upper and lower extremities were extracted from the Rikli & Jones Senior Fitness Test as study variables (strength is quantified according to the number of repetitions and flexibility is quantified in centimeters, through four simple tests) [16, 17]. On the other hand, the following variables were extracted from the SF-12 Questionnaire: the pain, the vitality, the physical and social functions, the general and mental health; the physical and emotional limitations; and two global scores on the physical and mental components of health (all the variables provided by this questionnaire are quantified in score points) [18-20].”

  1. Material and Methods: Some strategies to guarantee adherence were performed during the interventions?

The monitors had to maintain a careful, attentive and motivating attitude (proper to their position) at all times. In addition, after not attending two consecutive training sessions, the absent participant was called by telephone to find out the reason for her absence.

  1. Statistics analysis: The study has a 3x3 design (please include at the beginning of the methods part), i.e.: 3 groups and 3 times of measures for that reason I think that t-test is not the correct analysis since t-test is used to analyze the mean difference between maximum two groups or two times of measures. I suggest to authors perform a factorial repeated measures ANOVA because it is the ideal statistical test for the design of this study.

Both aspects have been consistently applied and corrected in the manuscript.

  1. Statistics analysis: Authors controlled other confounders?

We are aware that long-term follow-up can lead to confounders, but we had no way to carry out more control methods.

  1. Results: Please, use subsections to describe the results according to the activity groups, i.e. describe the results within the groups during the three assessments in separated subsections.

This modification has been made according to your instructions.

  1. Results: Please, report the adherence or % of participants' attendance.

In all the groups, it was found that all the women who had been leaving the program were also those who died. Although we did not establish a relationship between survival and the assigned activity group, the authors do reflect this in Table 1. This phenomenon is mentioned in Results and Discussion but without falling into speculation or statements that are beyond the scope of this research.

  1. Discussion: In the first paragraph of the discussion, the authors describe an expression about the rest of the population. Please, describe only your main findings related to your study sample.

The phrase you refer to has been removed.

  1. Discussion: Please, describe with more detail the results of the mentioned studies compared with your results (i.e. quantify and compare the improvements or detriments of the outcomes between the studies in the literature and your study).

Some sentences and references have been added in response to your request.

Once again, thank you very much for the time spent and the interest shown in this work; as well as in the positive evaluations you have given of it.

Receive a warm greeting,

The authors.

Round 2

Reviewer 1 Report

The authors have made substantial improvements in explaining the study design and the intervention programs themselves. The manuscript flows better. Thank you.

The issue remains, however, that this is not a double-blind RCT, at least not in the traditional meaning of the term and, therefore, misrepresents the study design in both the title and abstract. I appreciate that the team has taken considerable efforts to mask the evaluators during the testing phase of the study and the analysts and that the methods section now appropriately describes these efforts.   Here is a reference that defines the usual meaning of the term, double-blind: https://www.ncbi.nlm.nih.gov/books/NBK546641/

Therefore, I strongly recommend that the term ‘double-blind’ be removed from the title and in the abstract study design description. I would just call this a randomized controlled trial of three intervention modalities. In the methods then, the authors can detail the efforts to reduce bias by…blinding the evaluators and the analysts.

Author Response

Dear Editor and Reviewer of the International Journal of Environmental Research and Public Health:

Thank you very much for your suggestions and contributions to improve the quality of the manuscript. Following your indications, we respond, point by point, to your comments.

In the text, all the modified or added sentences have been written in red to facilitate the correction by the reviewers.

  1. The issue remains, however, that this is not a double-blind RCT, at least not in the traditional meaning of the term and, therefore, misrepresents the study design in both the title and abstract. I appreciate that the team has taken considerable efforts to mask the evaluators during the testing phase of the study and the analysts and that the methods section now appropriately describes these efforts. Therefore, I strongly recommend that the term ‘double-blind’ be removed from the title and in the abstract study design description. I would just call this a randomized controlled trial of three intervention modalities. In the methods then, the authors can detail the efforts to reduce bias by…blinding the evaluators and the analysts..

The authors have corrected the Title and the Abstract according to their advice.

Once again, thank you very much for the time spent and the interest shown in this work; as well as in the positive evaluations you have given of it.

Receive a warm greeting,

The authors.